# Contrasting effects of intracellular and extracellular human PCSK9 on inflammation, lipid alteration and cell death
Aram Ghalali[1], Fahd Alhamdan[2,3,4], Swapna Upadhyay[5], Koustav Ganguly[5], Kjell Larsson[5], Lena Palmberg[5] & Mizanur Rahman [5,6] ✉

Proprotein convertase subtilisin/kexin type 9 (PCSK9) is one of the major regulators of low-density lipoprotein receptor (LDLR). Information on role and regulation of PCSK9 in lung is very limited. Our study focuses on understanding the role and regulation of PCSK9 in the lung. PCSK9 levels are higher in Bronchoalveolar lavage fluid (BALF) of smokers with or without chronic obstructive pulmonary diseases (COPD) compared to BALF of nonsmokers. PCSK9-stimulated cells induce proinflammatory cytokines and activation of MAPKp38. PCSK9 transcripts are highly expressed in healthy individuals compared to COPD, pulmonary fibrosis or pulmonary systemic sclerosis. Cigarette smoke extract reduce PCSK9 levels in undifferentiated pulmonary bronchial epithelial cells (PBEC) but induce in differentiated PBEC. PCSK9 inhibition affect biological pathways, induces lipid peroxidation, and higher level of apoptosis in response to staurosporine. Our results suggest that higher levels of PCSK9 in BALF acts as an inflammatory marker. Furthermore, extracellular and intracellular PCSK9 play different roles.

PCSK9 (proprotein convertase subtilisin/kexin type 9) was first identified in 2003 and is the 9[th] member of proprotein convertase family[1]. As one of the key regulators of the low-density lipoprotein receptor (LDLR), it has been suggested as a target of potential lipid-lowering therapeutics[2,3]. Although the liver is the major source of PCSK9, expression in other tissues such as the kidney, small intestine, brain, heart, and blood vessels points out a possible role of the PCSK9 beyond LDLR degradation[4–7]. PCSK9 has been implicated in potentiating inflammatory responses. The pro-inflammatory properties of PCSK9 further indicates its multifaceted involvement in cellular function[6]. PCSK9 induced proinflammatory cytokines in macrophages[8] and PCSK9 inhibition enhanced immune checkpoint therapy against cancer[9]. In addition, circulating PCSK9 levels were significantly associated with cardiovascular diseases (CVD) among systemic lupus erythematosus (SLE) patients[10], wherein Ox-LDL-induced dendritic cell maturation was inhibited by PCSK9 inhibition[10]. Furthermore, we identified the role of PCSK9 in rheumatoid arthritis[11] and defined that patients with low levels of circulatory

PCSK9 exhibited a positive response to anti-TNF treatment, while those with higher circulatory PCSK9 levels did not respond to the therapy. Additionally, PCSK9-stimulated synoviocytes induced secretion of monocyte chemoattractant protein-1 (MCP-1). Moreover, PCSK9-induced MCP-1, TNF-α, and IL-1β secretion by macrophages in response to PCSK9 was effectively inhibited by anti-PCSK9 antibodies[11].

Involvement of PCSK9 has been investigated mainly in cardiovascular diseases (CVD). Respiratory disease and CVD are common cause of morbidity and mortality worldwide. These diseases often coexist, and the outcomes are worse than either of the conditions alone[12–14]. Although shared risk factors such as smoking, air pollution, poor diet, and occupational exposure are common for respiratory and cardiovascular diseases, the association between cardiovascular disease and respiratory conditions is not solely dependent on these risk factors[15], Indeed, there is a well-established relationship exists between the prevalence of CVD, airflow obstruction and mortality[13,14]. This highlights the complex interplay between respiratory and

[1]Vascular Biology Program, Boston Children Hospital, Harvard Medical school, Boston, MA, USA. [2]Department of Anesthesiology, Critical Care, and Pain Medicine, Cardiac Anesthesia Division, Boston Children's Hospital, Harvard Medical School, Boston, MA, USA. [3]Department of Immunology, Harvard Medical School, Boston, MA, USA. [4]Broad Institute of MIT and Harvard, Cambridge, MA, USA. [5]Unit of Integrative Toxicology, Institute of Environmental Medicine, Karolinska Insitutet, Stockholm, Sweden. [6]Department of Medicine, Brigham and Women's Hospital, Harvard Medical School, Boston, MA, USA. ✉e-mail: mizanur.rahman@ki.se

cardiovascular health, emphasizing the need for a comprehensive understanding of their interconnected mechanisms.

In the airway, the epithelial barrier is the first line of defense of the innate immune system that protects against environmental insults and infection. The bronchiolar epithelial cells in patients with respiratory disease, including COPD and asthma, exhibit a retarded wound repair process and proliferation of epithelial cells, both of which are correlated with the severity of the diseases and a potential cause of comorbidity. Although COPD patients have higher serum lipid levels, the association between serum lipid level and smoking is not well understood[16]. High levels of lipid in serum for patients with COPD has been suggested as a risk factor for cardiovascular complications such as atherosclerosis[16]. Recent studies suggest that alterations in lipid metabolic pathways contribute to pathogenesis of lung diseases, including fibrosis and COPD, plasma PCSK9 level was associated only with secondary outcomes of fewer intensive care unit free and ventilator free days[17]. Inhibition of PCSK9 reduced development of pulmonary arterial hypertension in vivo[18]. Expression of PCSK9 was identified in alveolar cell lineage A549[19] but function and biogenesis of PCSK9 have much remaining to be characterized. in airway epithelial cells. Here, we aimed to investigate whether PCSK9 exists in bronchial epithelial cells and identify the role of extracellular and intracellular PCSK9.

## Result

### BALF of smokers contain higher level of PCSK9

In our analysis, we identified PCSK9 in the bronchoalveolar lavage fluid (BALF) of both smokers and nonsmokers. Intriguingly, the levels of PCSK9 were markedly elevated in the BALF of smokers without COPD when compared to nonsmokers, as illustrated in Fig. 1a. This observation indicates a significant association between smoking and increased PCSK9 levels in the

BALF, highlighting a potential link between smoking and the regulation of PCSK9 expression in the respiratory environment.

### Extracellular PCSK9 induce proinflammatory cytokines in PBEC

In our exploration of the potential impact of extracellular PCSK9 from BALF on airway epithelial cells, PBEC on air-liquid interface (ALI) were stimulated with PCSK9. The results revealed a concentration-dependent increase in cytokines and chemokines, specifically IL-8 (Fig. 1b), TNF-α (Fig. 1c), IL-6 (Fig. 1d), and MMP9 (Fig. 1e), but IL-1beta was not detected in this context. Furthermore, the activation of phosphorylation in the MAPKp38 pathway was observed in response to PCSK9 (Fig. 1f). While MAPKp38 was activated, the phosphorylation of the NFkB p65 subunit remained unaffected in response to PCSK9, as presented in the Supplementary Fig. (Supplementary Fig. 1a). This response suggests a selective impact of PCSK9 on specific signaling pathways, shedding light on its potential role in modulating cellular responses in the airway environment. We further explored the role of extracellular PCSK9 on alveolar epithelial like cells (A549), revealing that PCSK9 induces MAPKp38 activation, as depicted in Supplementary Fig. (Supplementary Fig. 1b). To delve into inflammatory mechanism, we examined effects of extracellular PCSK9 on caspase-1 and TGF-B. Extracellular PCSK9 did not affect caspase1 or TGF-B significantly (Supplementary Fig. 1c).

### The expression of PCSK9 in both healthy and diseased conditions at a single cell- resolution

Lung harbor various cell types including resident and nonresident cell types, particularly in an inflammatory condition. We analyzed previously published multiple single-cell datasets to identify the potential major source of PCSK9 in BALF. This analysis identified 9 clusters of cells including AT I,

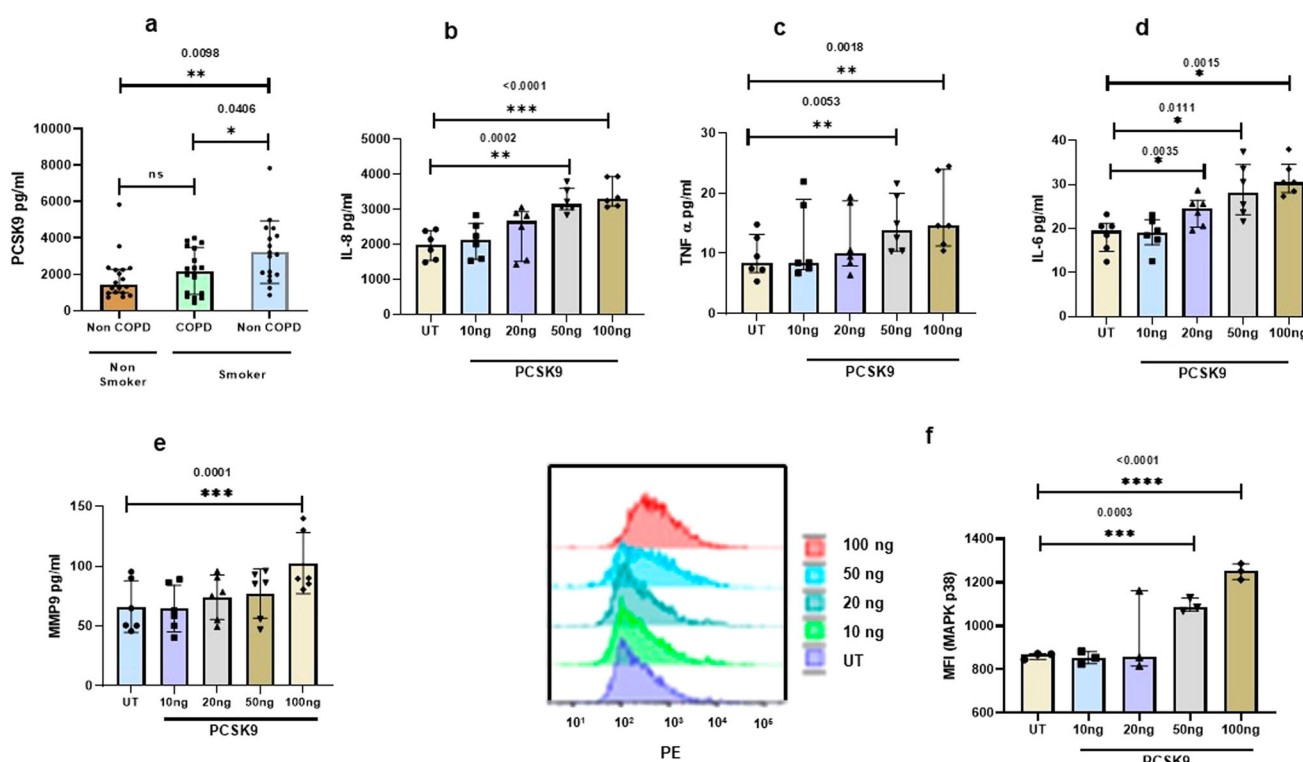

**Fig. 1 | Elevated PCSK9 levels in the BALF of Smokers and induction of inflammation by extracellular PCSK9. a** PCSK9 was measured from bronchoalveolar lavage fluid (BALF) of both smokers with (*n* = 18) or without chronic obstructive pulmonary disease (COPD) (*n* = 17) and non-smokers (*n* = 17). **b–d** Primary bronchial epithelial cells (PBEC) were cultured with PCSK9 or in untreated (UT) condition for 24 h. In response to stimulation with PCSK9, PBEC

induced production of pro-inflammatory cytokines, including IL-6, TNF-α, and the chemokine IL-8 (*n* = 6). **e** PCSK9d induced matrix metalloproteinase 9 (MMP9) expression, with a concentration-dependent effect (*n* = 6). **f** PCSK9 was identified as an inducer of phosphorylation of MAPKp38 (*n* = 3), with higher concentrations demonstrating a more pronounced effect. Error bars represent median interquartile range. *p*-value ≤ 0.05 was considered statistically significant.

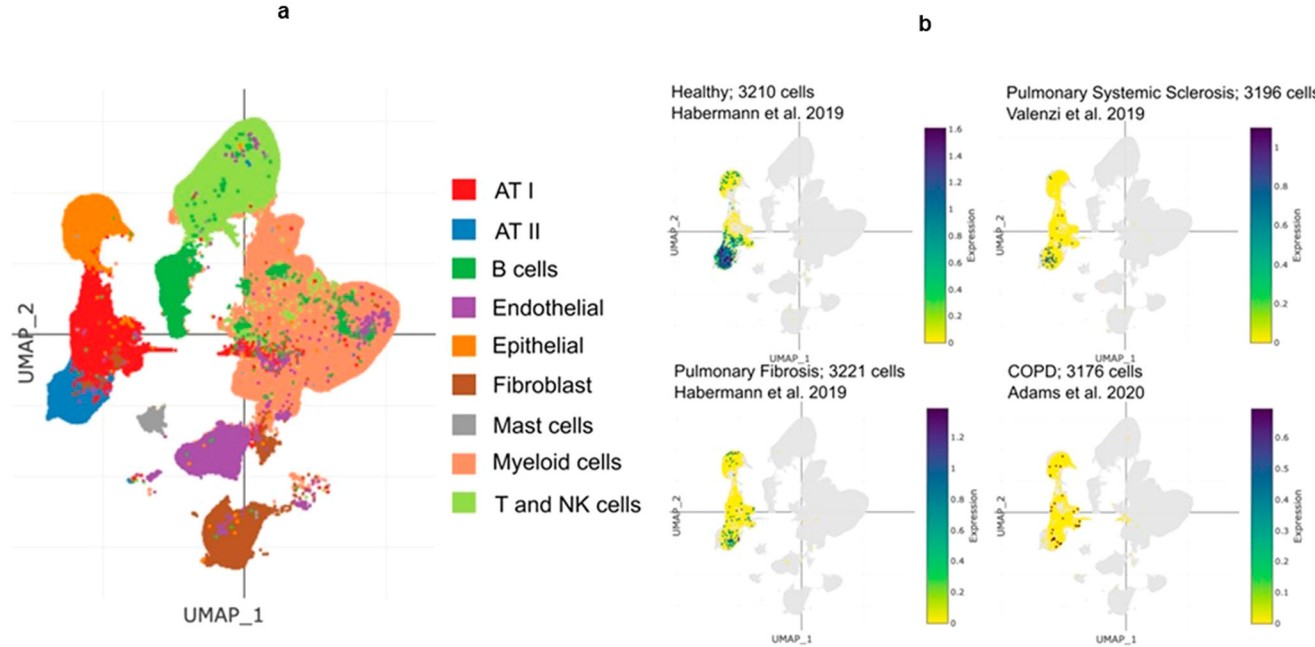

**Fig. 2 | Single cell analysis of lung homogenate. a**) Uniform Manifold Approximation and Projection (UMAP) representation of human lung single cells grouped into 9 distinct cell types. **b** Comparison of PCSK9 expression levels amongst healthy and different conditions of lung diseases (Healthy: 4 donors; Pulmonary Fibrosis: 6 donors; Pulmonary Systemic Sclerosis: 5 donors; COPD: 18 donors).

AT II, and bronchial epithelial cells (Fig. 2a). Focusing mainly on these three cell types, we were able to delineate the expression levels of PCSK9 in healthy and lung disease conditions by using similar number of cells in all conditions (~ 3200 cells) (Fig. 2b). The expression of PCSK9 transcripts was dramatically attenuated in AT II cells in pulmonary fibrosis (6 donors), pulmonary systemic sclerosis (5 donors), and chronic obstructive pulmonary disease (COPD) (18 donors) compared to healthy condition (4 donors). Particularly in COPD patients, PCSK9 levels were lower when contrasted with healthy individuals.

### PCSK9 is expressed by both PBEC and A549 cells

To ascertain the expression of PCSK9 in both upper and lower airways, quantitative measurement by ELISA was employed to detect PCSK9 levels in cell lysates of PBEC and A549 cells. Both PBEC and A549 cells express PCSK9 protein level (Fig. 3a). This expression level of PCSK9 protein level was further validated through western blotting (Fig. 3b) and no significant difference in the expression levels of PCSK9 between PBEC and A549 cells (Fig. 3a, b). Immunostaining provided additional insights, revealing that PCSK9 is present as both cytoplasmic and nuclear protein in both PBEC and A549 cells (Fig. 3c). Furthermore, PCSK9 level was higher in differentiated PBEC compared to non-differentiated PBEC (Fig. 3d).

### CSE induce level of PCSK9 expression in differentiated PBEC

To identify a potential connection of having higher level of PCSK9 in BALF, we demonstrated the effect of CSE on the level of PCSK9 in undifferentiated PEBC. CSE reduced PCSK9 protein level in undifferentiated PBEC (Fig. 3e, f). In CSE treated cells PCSK9 were located primarily in the cytoplasm (Fig. 3f). Extending the demonstration on A549 cells revealed a similar effect (Supplementary Fig. 2a). CSE induced necrosis in PBEC and caused release of intracellular molecules[20]. In addition, we revealed higher level of propidium iodide (PI) positive cells in CSE treated cells compared to untreated cells (Supplementary Fig. 2b). To identify whether PCSK9 released into the cell culture supernatant due to cell membrane being damaged by CSE, we measured and detected PCSK9 in the cell culture supernatant of CSE treated cells (Fig. 3g). Brefeldin which inhibits protein transport from the endoplasmic reticulum to the golgi, reduced CSE-induced PCSK9 minimally in compared to in the

absence of brefeldin (Fig. 3h). Interestingly PCSK9 minimally increased in differentiated PBEC in response to CSE (Fig. 3i).

### Intracellular PCSK9 deficiency affects cellular pathway

To investigate the impact of intracellular PCSK9 in PBEC, we transfected cells with PCSK9 siRNA to suppress PCSK9 expression. Inhibition of PCSK9 expression was confirmed by WB (Supplementary Fig. 3a). Initially we investigated inflammatory pathway MAPKp38, but PCSK9 inhibition did not affect this pathway significantly (Supplementary Fig. 3b). Subsequently, we conducted bulk RNA-Sequencing and focused on the top 2000 differentially expressed genes (DEGs) for further analysis of downstream biological pathways and processes (Fig. 4a). The differential expression of genes influenced various biological pathways and processes including upregulation of cholesterol biosynthesis, CD8/T cell receptor downstream pathway, positive regulation of extrinsic apoptotic signaling pathway, DNA methylation, and cyclooxygenase pathway (Fig. 4b, c). We additionally detected biological pathways and processes were affected by the down-regulated genes such as Cytochrome P450 pathway, P53 signaling pathway, Aryl hydrocarbon receptor signaling, and aryl hydrocarbon receptor signaling (Fig. 4b, c) Additionally, list of specific genes associated with some of these pathways are presented (Fig. 4d).

### PCSK9 regulates lipid metabolism and cell death

The silencing of PCSK9 had notable effects on cellular processes, as evidenced by induction of cholesterol synthesis pathway (Fig. 4d) but also the downregulation of cytochrome P450 pathway signaling (Fig. 4d). Based on bulk sequencing data, we aimed to validate cholesterol synthesis, apoptosis and cellular response against different components in PCSK9-suppressed cells. PCSK9 deficiency led to an increase in cholesterol levels in PBEC, as confirmed by measuring lipid by flow cytometry analysis (Fig. 5a). Smoking induces lipid peroxidation[21]; we identified that PCSK9 inhibition induced lipid peroxidation at a higher level in CSE-treated cells (Fig. 5b, c). Additionally, we observed that cell proliferation was reduced in PCSK9-silenced cell compared to control siRNA transfected cells. We investigated whether cell cycle is affected and revealed that PCSK9 inhibition affected the cell cycle (Fig. 5d). Although S phase was induced minimally but G2-M phase was affected in PCSK9 siRNA transfected cells compared to control siRNA

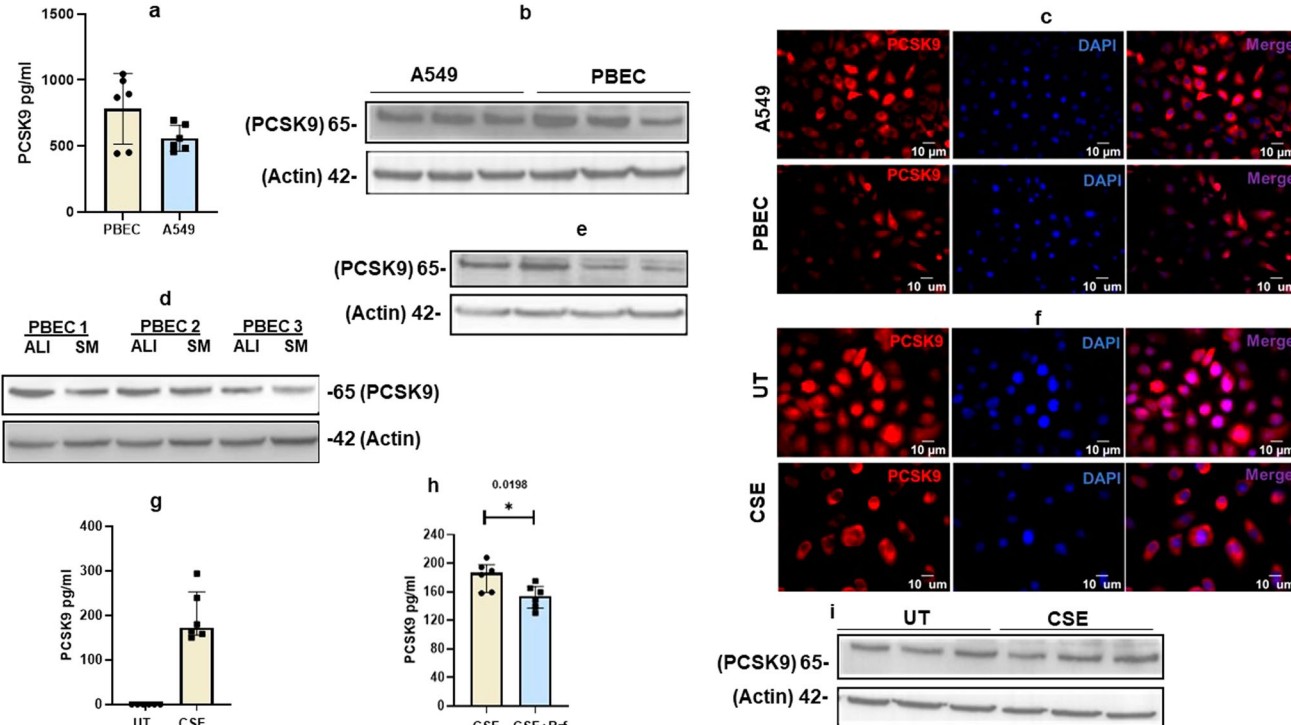

**Fig. 3 | Influence of cigarette smoke extract (CSE) on PCSK9 expression levels.**
**a** PCSK9 levels were quantified from cell lysates of A549 cell ($n = 6$) and PBEC
($n = 6$) by ELISA. **b** Western blot from the cells lysates of A549 cells and PBEC.
**c** In addition, immunostaining was performed to visualize the microscopic images
for the PCSK9, scale bar 10 µm. Both PBEC and A549 cells expressed PCSK9 at the
protein level. **d** Differentiated PBEC on air liquid interface (ALI) exhibited higher
levels of PCSK9 protein compared to undifferentiated PBEC in submerged culture
(SM). **e** In compared to untreated (UT) condition, treatment with cigarette smoke

extract (CSE) resulted in a reduction of endogenous PCSK9 protein levels in
undifferentiated PBEC. **f, g** Undifferentiated PBEC, upon CSE treatment, released
PCSK9 into the cell culture ($n = 6$). Notably, this CSE-induced PCSK9 release was
inhibited by brefeldin ($n = 6$). **h** In CSE-treated PBEC, PCSK9 was primarily located
in the cytoplasm, 10 µm scale bar was used for microscopic images **i** Minimal
increase in PCSK9 was observed in differentiated PBEC in response to CSE. Error
bars represent median interquartile range. *p*-value ≤ 0.05 was considered statistically
significant.

transfected cells (Fig. 5d). Cell cycle alteration is connected to apoptosis or
cell death. Although, we did not observe cell death in PCSK9 siRNA
transfected cells, higher level of cellular death was observed in response to
CSE in PCSK9-siRNA transfected cells compared to control siRNA trans-
fected cells (Fig. 5e). Further experiments demonstrated the effects of drugs
in PCSK9-silenced cells. In the absence of PCSK9, cells did not undergo
apoptosis; however, staurosporine induced higher level of apoptosis in
PCSK9-silenced cells compared to control siRNA transfected cells (Fig. 5f).
Additionally, active caspase 3 was induced (Fig. 5g), although the cleavage of
PARP was not significantly affected (Supplementary Fig: 4). We investigated
whether extracellular PCSK9 has any effect on cellular growth. We did not
observe any significant impact on cellular proliferation when cells stimu-
lated with extracellular PCSK9 (Supplementary Fig. 5).

## Discussion
While PCSK9 has been implicated in various diseases, and its inhibition has
been proposed as a therapeutic target, the biogenesis and its role has not
been adequately explored, especially in connection with the airways. This
study provides valuable insights into a novel aspect of PCSK9 biology,
particularly within the context of lung cells. We show that extracellular and
intracellular PCSK9 in lung epithelial cells have differential impacts on
inflammation, metabolic activity, cellular growth, and proliferation. We
reported, for the first time, the presence of PCSK9 in bronchoalveolar lavage
fluid (BALF), and higher levels of PCSK9 in smokers, particularly smokers
without COPD. Surprisingly, level of PCSK9 was lower in the smokers with
COPD compared to the nonsmokers without COPD. We speculate that
smoking-induce cellular damage releases intracellular PCSK9 but in COPD
condition it is possible that PCSK9 expression is suppressed, leading to
PCSK9 release at lower level compared to smokers without COPD. Our

findings revealed that extracellular PCSK9 induces inflammatory cytokines
and activates inflammatory pathways, suggesting that elevated levels of
PCSK9 in BALF trigger inflammation in the airways and potentially con-
tribute to inflammatory exacerbations in disease conditions. MMP9 is
implicated in fibrosis[22] and severity of COPD[23]. In our investigation,
PCSK9-stimulated PBEC induced MMP9, suggesting a potential cause of
tissue injury and exacerbation in COPD. We determined that there are
higher levels of PCSK9 transcripts in the healthy individuals compared to
patients' groups, suggesting again that disease condition affecting PCSK9
expression. Furthermore, low level of intracellular PCSK9 may be a risk
factor for such lung diseases development. The observation that the dif-
ferentiation state of these cells may contribute to PCSK9 expression is
intriguing. Airway basal cells in certain circumstances differentiate into
different cell types including goblet cells[24]. Cigarette smoke is well known for
inducing goblet cell differentiation and production of mucus[25]. In disease
conditions including chronic inflammation and COPD, goblet cell hyper-
plasia is common. Higher level of PCSK9 expression in the differentiated
PEBC compared to undifferentiation PBEC, adds another layer of com-
plexity to this dynamic process but indicating that some other differentiated
PBEC express higher level of PCSK9 compared to goblet cells. The obser-
vation that undifferentiated PBEC reduced PCSK9 in response to cigarette
smoke extract (CSE) and exhibited cytotoxic effects against CSE exposure is
noteworthy. The nuclear and cytoplasmic presence of PCSK9, along with
the indication that its translocation may occur, opens avenues for further
investigation into the dynamics of PCSK9 localization and a potential role of
this protein in cellular responses. The detection of PCSK9 in the cell culture
supernatant of CSE-treated undifferentiated cells, along with observation of
damaged membranes, provides a compelling explanation for the higher
levels of PCSK9 in smokers BALF compared to healthy condition. These

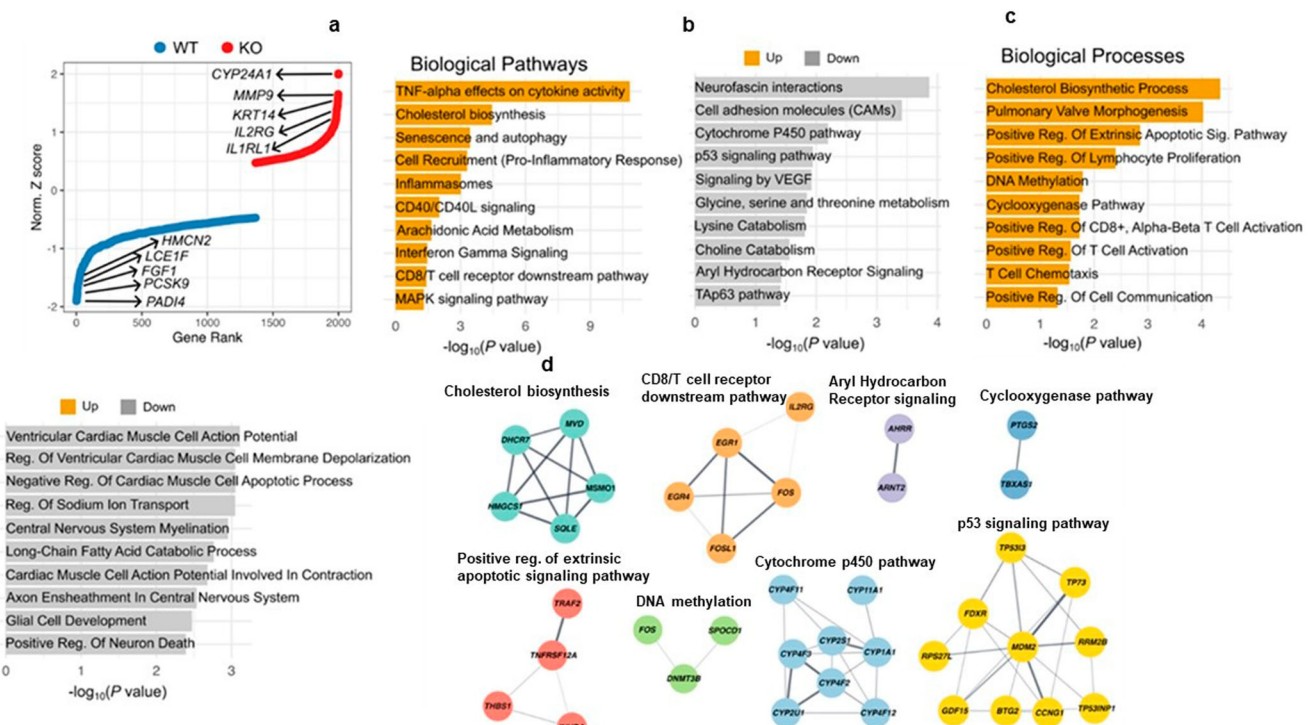

**Fig. 4 | Influence of PCSK9 inhibition on cellular pathways and processes.** Bulk RNA sequencing analysis from control siRNA and PCSK9 siRNA transfected PBEC. **a** Endogenous inhibition of PCSK9 showed alterations in the expression of approximately 2000 genes, both upregulated and downregulated genes. **b, c** The impact of PCSK9 inhibition extended to crucial biological pathways and processes, demonstrating an influence on cellular functions. 10 upregulated and 10 down-regulated crucial pathways or biological process were presented. **d** Gene co-expression network of affected genes encompassed pathways and processes vital to cellular function, such as cholesterol synthesis, COX, apoptosis, ARH (Aryl hydrocarbon receptor), and the CYP450 pathway.

findings suggest that cellular damage contributes to the release of intracellular PCSK9 into the BALF. The differentiation state of PBEC appears to play a crucial role as differentiated membrane integrity remains unaffected by CSE and induced PCSK9 expression, potentially as a protective response. The observation indicates that airway epithelial cells respond to pollutants such as cigarette smoke by inducing PCSK9 expression. This has implications for understanding the broader impact of PCSK9 in the context of airway responses to external stressors. PCSK9 targeting has been suggested in several studies[9,26–31]. A recent clinical study named IMPACT-SIRIO 5 (Impact of PCSK9 Inhibition on Clinical Outcome in Patients During the Inflammatory Stage of the COVID-19, compared the patients with severe COVID-19 treated with PCSK9 inhibitor had significantly lower rates of death or need for intubation within 30 days compared to those who received the placebo[29]. Inflammatory cytokines in serum decreased more in patients treated with the PCSK9 inhibitor compared to those on placebo. Additionally, the PCSK9 inhibitor compared to placebo, reduced mortality in patients who had higher level of baseline IL-6 levels. These findings suggest that PCSK9 regulate inflammation in lung and thus PCSK9 inhibition may be beneficial in reducing inflammation and improving outcomes in severe COVID-19 cases, especially in patients with high levels of inflammation. However, in addition to monoclonal antibody-based inhibition of PCSK9, inhibition PCSK9 by RNA interference claimed a safer alternative for PCSK9 inhibition[31]. RNA interference based PCSK9 inhibition has potential impact to inhibit intracellular and extracellular PCSK9. However, the role of intracellular PCSK9 remains an intriguing aspect that requires further investigation in health and in different diseases. In this study, the differential regulation of genes and signaling pathways elucidated by inhibiting intracellular PCSK9 suggests involvement of PCSK9 in cholesterol biosynthesis, inflammation, and signaling against xenobiotics. The accumulation of intracellular lipids upon PCSK9 inhibition highlights its role in cholesterol synthesis and accumulation. Our findings elucidating the understanding of PCSK9's impact on cholesterol efflux are particularly interesting. While circulating or extracellular PCSK9 is known to reduce cholesterol efflux, our results suggest that intracellular PCSK9 may enhance cholesterol efflux, presenting a different perspective on PCSK9's influence on lipid metabolism. Lipid homeostasis is indeed crucial for overall physiological process, and the interplay between extracellular uptake and intracellular lipid dynamics plays a pivotal role in cellular function and phenotype[32,33]. A study has identified a positive correlation between PCSK9 and fibrosis in non-alcoholic fatty liver disease[34]. A recent study revealed that cholesterol depletion reduced the phosphorylation of basal Smad2/3, whereas cholesterol enrichment enhanced TGF-β1-mediated pSmad2/3 formation[35]. MMP9 induced TGF beta activation[36], but we did not see any impact on TGF beta levels by extracellular PCSK9. However, intracellular PCSK9 inhibition-caused cholesterol enrichment potentially triggers TGF beta activation. Both aryl hydrocarbon receptor (AhR) and cytochrome P450 (CYP450) play crucial roles in various physiological and toxicological signaling pathways. In our study, the inhibition of PCSK9 resulted in the downregulation of both AhR and CYP450 pathways, suggesting that PCSK9 protect cells from xenobiotics. The crosstalk between AhR and CYP450 enzymes responds to different compounds, including environmental pollutants, toxins, and drugs. The diverse functions of CYP450 enzymes in liver diseases, cardiovascular diseases, and cancer underscore the importance of understanding their activities in different cancer types.

Some CYP450 forms are selective for tumor cells, potentially contributing to drug resistance[37]. AhR and CYP450 regulate cancer cell proliferation and development, self-renewal and chemoresistance through inhibition of the PTEN and activation of β-Catenin and Akt pathways[38]. Whether to suppress or enhance CYP450 activities depends on the specific cancer type and the metabolites produced by these enzymes. Variations in cytochrome P450 expression and activity can significantly influence individual responses to chemotherapy, potentially affecting treatment outcomes. Identifying the specific enzymes affected by intracellular PCSK9 could provide valuable guidance on whether suppressing PCSK9-mediated

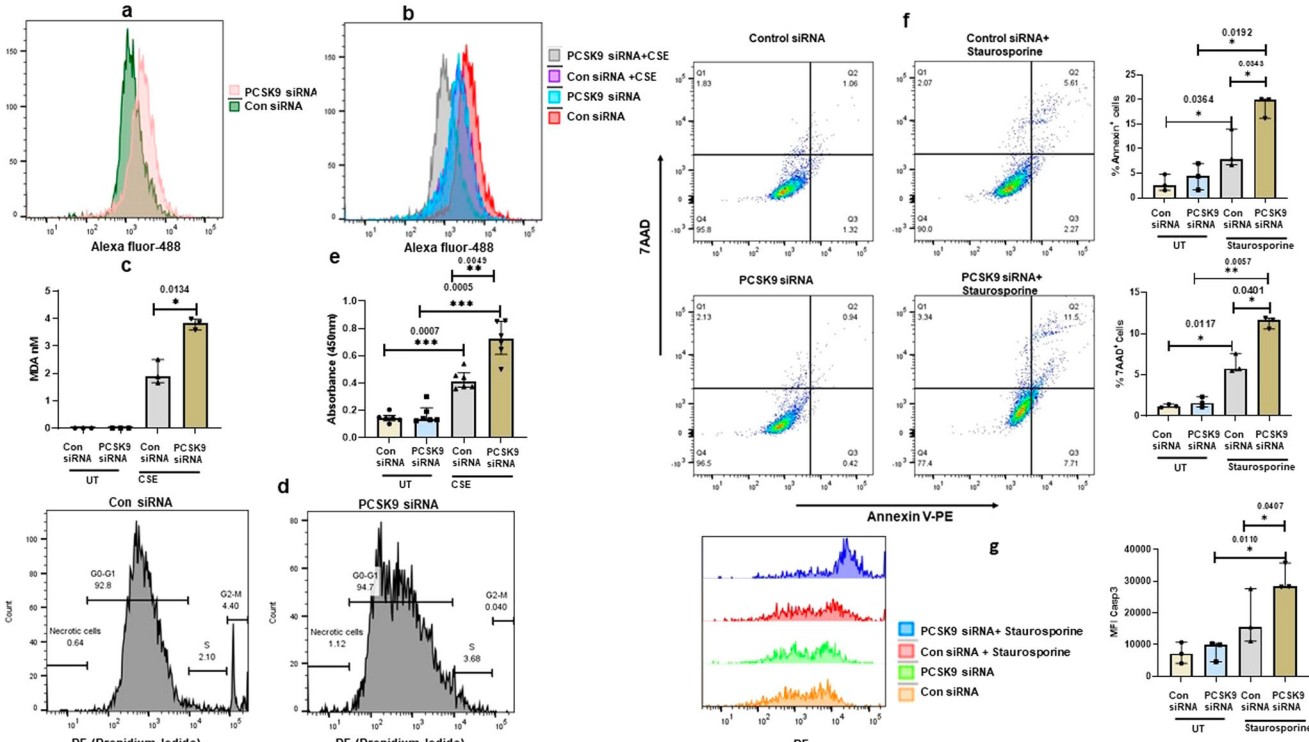

**Fig. 5 | Impact of PCSK9 Inhibition on Lipid Accumulation, Lipid Peroxidation, and Cellular growth. a** Lipid level was measure by lipid accumulation specific bodipy reagent. Inhibition of PCSK9 was associated with an increase in lipid accumulation. **b, c** Lipid peroxidation was measured by lipid peroxidation specific bodipy and lipid peroxidation product MDA (*n* = 3). In compared to untreated (UT), cigarette smoke extract (CSE) induced a higher level of lipid peroxidation in cells in the PCSK9-suppressed PBEC. **d** Cell cycle was analyzed by propidium iodide staining. PCSK9 inhibition affected cell cycle, primarily affecting the G2/M phase. **e** LDH assay was performed to determine cell death. CSE-induced cell death was notably higher in PCSK9-silenced cells compared to control siRNA transfected cells (*n* = 6). **f,g** In response to the staurosporine, PCSK9 inhibition induced apoptosis and an increase in caspase 3 activation (*n* = 3). Error bars represent median inter-quartile range. *p*-value ≤ 0.05 was considered statistically significant.

CYP450 inhibition is beneficial for cancer treatment. However, it remains to be elucidated whether intracellular PCSK9 enhances the activation or inactivation of any drugs. In the absence of intracellular PCSK9, cells did not induce apoptosis significantly, but displayed altered cell division, aligning with the findings, and explaining the mechanism of targeting tumorigenic cells. Endogenous PCSK9-supprssed cells leads to higher levels of apoptosis compared to control siRNA transfected cells. Staurosporine is a protein kinase inhibitor known for its cancer cell inhibitory function. This finding suggests a potential role for endogenous PCSK9 inhibition in cancer therapy. These findings shed light on the intricate interplay between PCSK9, lipid metabolism, and cellular responses, offering valuable insights into potential therapeutic strategies. Extracellular and intracellular PCSK9 may play distinct roles in health and disease conditions. Lipid metabolism is often altered in tumor cells, characterized by the upregulation of lipid peroxidation, lipid synthesis, and storage of lipids, which aids cells in nutrient-deficient conditions[39]. However, lipid peroxidation in certain circumstances induce ferroptosis. Our finding upregulation of lipid peroxidation suggest that PCSK9 inhibition induce ferroptosis when antioxidant especially GPx4 is compromised. PCSK9 inhibition reduced liver cell proliferation[40], PCSK9 induced cancer cell growth by inhibiting apoptosis[41], which aligns with our findings, inhibition of cell cycle and induction of apoptosis in response to drugs. Limitation of this study include a small number of BALF samples, therefore future investigation with a higher number of clinical samples is warranted.

Our findings suggest that extracellular and intracellular PCSK9 have differential roles in cellular homeostasis of airway epithelial cells. Future investigations into identifying the origin of PCSK9 in various pathological conditions for tissue-specific targeting of PCSK9 should be considered. The link between PCSK9, xenobiotics, and airway inflammation may have implications for understanding and potentially treating diseases with an inflammatory component in the respiratory system. Extracellular stimulation by PCSK9 on lung epithelial cell and effect of intracellular PCSK9 suppression were presented in a hypothetical model (Fig. 6). In the context of lipid synthesis or storage, our study indicates that only intracellular targeting of PCSK9 may not be a sufficient strategy; rather, inhibition of intracellular PCSK9 potentially enhances the effect of drugs, therefore pointing towards a combination therapy. We suggest that higher levels of PCSK9 in BALF are to be considered as a risk factor for inflammatory exacerbation in lung diseases including fibrosis and COPD but intracellular PCSK9 may protect against development of lung disease. Additionally, higher levels of PCSK9 in BALF trigger pulmonary hypertension and cardiovascular complication. Role of intracellular PCSK9 in cellular homeostasis highlighting that targeting these forms of PCSK9 should be tailored to specific diseases. The need for further investigations to comprehensively understand the roles of extracellular and intracellular PCSK9 in various health and disease conditions is emphasized. This approach aims to contribute to a more understanding of PCSK9's involvement in lung-related pathologies and may guide future therapeutic strategies against different diseases.

## Methods
### Bronchoalveolar lavage
Total 60 individuals, non-smokers (20), current smokers (20) and smokers with COPD (20) were recruited by advertisement in daily press. Inclusion criteria for the COPD group required post-bronchodilator FEV1/FVC ratio < 0.70, post-bronchodilator FEV1 between 40 and 70% of predicted value, and arterial oxygen saturation (SaO₂) > 90%. Smokers without COPD were included who exhibited post-bronchodilator FEV1/FVC > 0.70 and

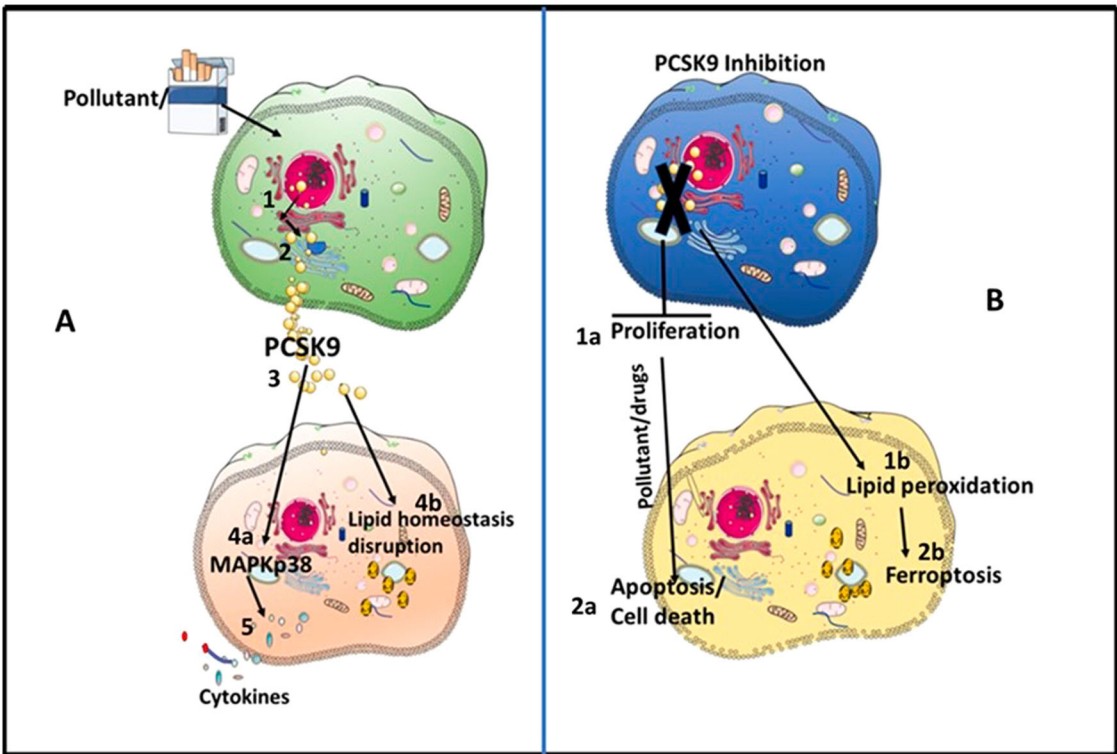

**Fig. 6 | Hypothetical model on cellular effects by PCSK9 stimulation and suppression of endogenous PCSK9. A** In response to xenobiotics PCSK9 transport from nucleus to Golgi through endoplasmic reticulum and transport vacuoles (1-2). Damaged cell membranes allow the PCSK9 release outside the cells (3) and stimulate other cells to induce MAPK activation and release cytokines (4a). Additionally, PCSK9 alter lipid hemostasis (4b). **B** In absence of PCSK9, cell cycle alteration results reduced proliferation (1) and in that condition exposure to xenobiotic induces cell death (2) Additionally, in absence of PCSK9 lipid peroxidation (1b) may induce ferroptosis (2b). The image was generated by using smart servier medical art tools. (https://smart.servier.com/citation-sharing). The elements were remixed and incorporated with power point elements.

FEV1 > 80% of predicted value. The control group consisted of healthy nonsmokers with normal spirometry. The characteristic of the individuals was described in an earlier study[42]. All participants provided informed consent, and the study was approved by the ethics committee at the Karolinska Institutet, Stockholm, Sweden (Dnr 2005/733-31/1-4). All ethical regulations in relevant to human research participants were followed. Patients' recruitment and collection were conducted in 2 years' time period and all the bronchoalveolar lavage fluid (BALF) was carefully obtained from the individuals the collected BALF samples were promptly preserved at −80 °C to ensure optimal conditions for subsequent analysis. In the current study, the BALF was used to measure level of PCSK9 form nonsmokers (17) smokers without COPD (17) and smoker with COPD (18).

## Cell culture
Pulmonary bronchial epithelial cells (PBEC) were collected from healthy part of bronchial tissues from donors in connection with lobectomy. All the donors gave written informed consent. The protocol was approved by the Swedish Ethical Review Authority (reference number: 99–357; approved on 10th January 2000). For experimental purposes, the cells were used passage 3-5. In addition, one batch of PBEC was used from commercially available from Lonza. The PBEC were cultured on air-liquid interface (ALI) or in submerged conditions. The PBEC were cultured on ALI according to the established protocol[43,44]. In brief, PBEC ($N = 3$) in submerged culture were grown and expanded using pneumocult expansion media (Stemcell Technologies, UK, cat# 05008). For ALI culture on transwell, maintenance was conducted using pneumocult ALI media (Stemcell Technologies, UK, cat# 05001). A549 cells were cultured in DMEM media supplemented with 10% FBS and 1% penicillin-streptomycin. The cells were treated with 5 µg/ml of cigarette smoke extract (CSE) (Fisher Scientific, US, product of Murty Pharmaceuticals, cat# NC1560725) or various concentrations of recombinant PCSK9. The recombinant PCSK9, expressed in HEK 293 cells, was purchased commercially (Sigma Aldrich, US, cat# SRP6285). Limulus amebocyte lysate assay was performed to ensure the protein free of endotoxin. Additionally, any impurities such as absence of RNA or DNA was ensured by nano drop absorbance spectrum at 260 nm or 280 nm. To suppress expression of PCSK9, PBEC were transfected with PCSK9 siRNA (Santa Cruz, Germany, cat# sc4548-2). Cells were transfected with siRNA plasmid by lipofectamine (Santa Cruz, Germany, cat# sc-29528). After 24 h of transfection, cell culture media was replaced with fresh media and cells were collected after 72 h for PCSK9 expression analysis by WB. Apoptosis was induced by Staurosporine (Sigma Aldrich, cat# S6942) according to the earlier established protocol[45]. In essence, PBEC were exposed to 100 nM Staurosporine for 24 h.

## ELISA
BALF was thawed at room temperature and subsequently resuspended in the solution containing 0.5% bovine serum albumin (BSA) in phosphate-buffered saline (PBS). To keep the measurement value within the kit detection limit, BALF were diluted 5 times with the 0.5% BSA containing PBS. Following the manufacturer's protocol (Botecnchne, UK), the PCSK9 Duoset ELISA kit (Biotechne, UK, cat# DY3888) was used to quantify the levels of PCSK9 in both cell lysates and BALF. Cell culture supernatant was collected after 24 h, determined by the specific requirements of each experiment. The levels of TNF-α, IL-6, IL-8, MMP9, and IL-1β were measured by ELISA Duoset kits for cytokines and chemokines (Biotecnhe, UK, cat# DY210, DY206, DY208, DY911 and DY201 IL-1β).

## Bulk RNA sequencing and bioinformatic analysis
Total RNA was extracted by RNA extraction mini kit (Qiagen, Germany). cDNA generation, library amplification and sequencing on NGS platform

was generated by novogene. Sequencing read depth was 20 million. Sequencing data was processed with the Galaxy platform. Quality of sequencing reads was assessed by FastQC v0.72. Reads were further aligned to the human reference genome (GRCh38/hg38) with RNA STAR v2.7.8a. Mapped reads were counted with feature. Counts v2.0.1, was followed by K-means clustering analysis to determine the differentially expressed genes (DEGs). Functional analysis including biological pathways and processes were curated with the implementation of different databases such as Reactome, Biological process (Gene Ontology), and NCATS BioPlanet48-50 by utilizing enrichR package (Version 3.2).

### Single-cell analysis
Single-cell RNA-Seq analysis was performed on the Single Cell PORTAL of the Broad Institute of MIT and Harvard with default settings. The data was retrieved from the study of Fabre et al.[46]. Cell clusters were annotated to their corresponding cell types and then filtered out according to cell types of interest and cell number. The datasets and number of donors used in this analysis were as follows: Healthy: Habermann et al. 4 donors; Pulmonary Fibrosis: Habermann et al. 6 donors[47], GEO accession number GSE135893; Pulmonary Systemic Sclerosis: Valenzi et al. 5 donors[48], GEO accession number GSE128169; COPD: Adams et al. 18 donors[49], GEO accession number GSE136831)

### Western blot
Cell lysates were prepared from PBEC or A549 cells by RIPA buffer with 1% protease inhibitor. Concentration of total protein was measured by BCA assay (Thermofisher, US, cat#23225) and 30 or 40 μg of protein of each sample was run in 12% gel and transferred onto PVDF membrane. The membranes were incubated with rat anti-human PCSK9 antibodies(cell signaling, US, cat# 85813, 1:1000 dilution), anti-Caspase1(Novus Biologicals, CO, US, cat# NBP1-45433, 1:1000 dilution), Anti-human TGF-beta1(Cell Signaling, US, cat# 3711, 1:1000 dilution), and mouse anti-beta actin (Sigma Aldrich, US, cat# A2228, 1:10,000 dilution) or GAPDH (Cell Signaling, cat# 2118 S) antibodies (1:10,000 dilution). Peroxidase conjugated secondary goat anti-rabbit (Thermo-Fisher, US, 1:10000 dilution) and rabbit anti-mouse (Sigma Aldrich, US, cat# A9044, 1:10000 dilution) antibodies were used to acquire the image. Membranes were developed by using substrate (Thermofisher, US, cat# 23225).

### LDH assay
Cells were transfected with PCSK9 siRNA or control siRNA as above. Transfected cells were treated with CSE, and cell culture supernatant was collected after 48 h of incubation. Cell viability was measured by LDH assay according to the manufacturer protocol (Thermofisher, US, cat# C20300).

### Flow cytometry
Flow cytometry was used for various purposes. In accordance with established protocol, cells were permeabilized by 0.1% saponin and stained with anti- phosphorylation of MAPKp38 (BD Biosciecne, cat#562065) or NFkB p65 antibodies (BD Bioscience, US, cat# 562065). To detect apoptosis, cells were stained with Annexin V and 7 AAD kit (BD Bioscience, US, cat# 559763). According to the manufacturer instruction, cells were incubated with Bodipy (Thermofisher, US, cat# D3922) reagent to quantify intracellular lipid droplet accumulation or lipid peroxidation specific Bodipy (Sigma Aldrich US, cat# D3861). To identify levels of active caspase 3 or PARP, cells were permeabilized with 0.1% saponin and intracellular staining was performed by antibodies against active caspase 3 or PARP (BD Bioscience, cat#570185). All the samples after staining and subsequent washing steps, were quantified by flow cytometry (LSR Fortessa). Raw data from flow cytometry was analyzed by flow Jo software, version 10. Flow cytometry analyzed data was presented as mean fluorescence intensity or percentage of cell population. Gating strategy of the flow cytometry was presented in the supplementary data file 2.

### Microscopy
Cells were grown on poly D lysin coated glass slides (Thermo fisher, US, cat# A3890401), washed with PBS, and fixed with a solution containing 3% paraformaldehyde and 0.25% glutaraldehyde. Following these preparatory steps, cells were incubated with 0.2% Triton and subsequent treatment with 3% BSA. Post-blocking with BSA, cells were incubated with anti-PCSK9 antibodies (Cell signaling, cat# 85813) and followed by incubation with Alexa Fluor 594 conjugated anti-rabbit secondary antibodies (Thermofisher, USA, cat# A-11012). The acquisition of microscopic images was acquired by LSM 800 Zeiss at 40X.

### Statistics and reproducibility
Statistical analysis for BALF samples were analyzed by one-way ANOVA followed by T test between each group. Experiments with A549 were conducted using three different passages and experiments with PBEC were conducted with cells from 3 individuals. The results were graphically represented in bar diagrams, expressing the median interquartile ranges, statistical analysis was performed by one-way ANOVA and followed 2-tailed T test within each condition. Statistical significance was considered with a $P$-value ≤ 0.05. The statistical analysis for the entire dataset was conducted using GraphPad Prism 8.30 software.

### Reporting summary
Further information on research design is available in the Nature Portfolio Reporting Summary linked to this article.

### Data availability
Raw data for bulk RNA sequencing has been deposited on Zenodo repository. Figure 4 was generated from processed data in the excel sheet (File name -mapped reads). https://doi.org/10.5281/zenodo.12820595. Western blot raw image files are available in supplementary section as a file name supplementary data file 1. Gating strategy for flow cytometry is available in supplementary section in the file ´supplementary data file 2´. All the numerical values for the Figs. 1, 3a, 3g, 3h, 5c, 5e, 5f, 5g, Supplementary Figs. 3b, and 4 are available in the supplementary data file 3 (excel sheet). All other data are available from corresponding author on reasonable request.

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

## Acknowledgements
Swedish Heart and Lung Foundation. Smart servier medical art. https://smart.servier.com/citation-sharing.

## Author contributions
M.R. conceived and designed the study, performed experiments, and wrote the manuscript. A.G., K.G., and S.U. performed experiments and contributed in study design and manuscript writing. F.A. conducted bioinformatic analysis and contributed in manuscript preparation. K.L. and L.P. characterized the clinical samples and designed study with BALF and contributed in manuscript writing.

## Funding

## Competing interests
The authors declare no competing interests.

## Ethical approval
All participants provided informed consent, and the study was approved by the ethics committee at the Karolinska Institutet, Stockholm, Sweden (Dnr 2005/733-31/1-4 and 99–357). All ethical regulations were followed.
