## [Peer Review File · Communications Biology]

Reviewers' comments:

Reviewer #1 (Remarks to the Author):

Ghalali et al sought to investigate the measurement of PCSK9 and its effect on several pathways bronchioalveolar lavage fluid (BALF). Experiments involved measurement of PCSK9 in bronchioalveolar lavage fluid (BALF), 32 single cell sequencing analysis, and exposure of lung epithelial cells to PCSK9, cigarette smoke 33 extract (CSE), and staurosporine. Their main findings are that PCSK9 stimulated cells induced proinflammatory cytokines and activation of MAPKp38.

The study might be of interest but the authors should address the following comments:

1) despite the large body of experiments, the authors lacked of clear explanation of their results. Could you please put in context your findings of most recent evidence. For instance, you can corroborate your results discussing the study named IMPACT SIRIO published in JACC which found a remarkable antiinflammatory action of PCSK9 inhibition in severe COVID, in turn leading to signal of improved clinical events including mortality in the population with higher baseline inflammatory risk and pulmonary inflammatory disease.

2) The authors' analyses should be more in-depth focused on statistical methodology to clarify significance and comparisons.

The author never displayed numerical values and is unclear to interpret and contrast results among experiments

3) Could the authors delve into the effect on inflammation showing additional results on biomarkers of inflammation by PCSK9

4) is unclear how to interpret the effect on apoptosis induced by PCSK9. Similarly, the effect on CYP450 what clinical implications may have, you should clarify this.

Reviewer #2 (Remarks to the Author):

In their study, "Contrasting Effects of Endogenous and Extracellular PCSK9 on Inflammation, Lipid Alteration and Cell Death," Ghalali and colleagues focus on understanding the role and regulation of PCSK9 in the lung. PCSK9 regulates the low-density lipoprotein receptor (LDLR) and has been deployed as a lipid-lowering therapy for those at risk of cardiovascular disease. Although the liver is the major source of PCSK9, expression in other tissues such as the kidney, small intestine, brain, heart, and blood vessels suggests a role of PCSK9 beyond LDLR-mediated lipoprotein regulation (Nature Reviews Endocrinology; 2019. doi.org/10.1038/s41574-018-0110-5). As the authors nicely outline, one of the dimensions that PCSK9 could be working through is regulating inflammatory responses.

The authors found that PCSK9 levels were higher in the BALF of smokers with or without chronic obstructive pulmonary diseases (COPD) compared to nonsmokers. PCSK9-stimulated cells induced proinflammatory cytokines and activation of MAPKp38. PCSK9 transcripts were highly expressed in healthy individuals compared to those with COPD, pulmonary fibrosis, or pulmonary systemic sclerosis. PCSK9 inhibition affected biological pathways, inducing lipid peroxidation and a higher level of apoptosis in response to staurosporine. The authors conclude that higher levels of PCSK9 in BALF act as an inflammatory marker and that extracellular and endogenous PCSK9 have different functional roles. The study has several strengths, including the significance/impact (understanding the diverse roles of PCSK9 in health and disease), the multidimensional phenotyping (human samples to cell models), and suggesting potentially distinct functional roles between intra- and extracellular PCSK9 levels. Although the significance of the study is high, a few weaknesses limit the enthusiasm for the manuscript as written:

- Please include more specifics on how subjects were identified and recruited (and the time period they were recruited over).
- [Page 8; Figure 1F] Please clarify the source of the exogenous PCSK9 used in experimentation and if it was tested for LPS and other inflammatory impurities
- The findings pertaining to distinct functions of intra- and extracellular PCSK9 have therapeutic implications that can be further expanded on in the discussion. Specifically, some of the newer PCSK9 RNA therapeutics could have effects on both intracellular and extracellular PCSK9 pathways compared with PCSK9 antibodies.
- Minor comments:
 - o Consider clarifying wording surrounding “intracellular” and “endogenous” PCSK9. Endogenous can be intra or extracellular, and the authors appear to be making the distinction between intracellular and extracellular PCSK9.
 - o As PCSK9 inhibitors in CVD have been thought to be working primarily through LDL-C regulation, in line with the author's premise, consider citing a recent PCSK9 inhibitor trial in ASCVD demonstrating modulation of inflammation in ASCVD (DOI:<https://doi.org/10.1016/j.atherosclerosis.2024.117529>).
 - o [Page 4, Cell culture methods] Please clarify: “In essence, PBEC were exposed to 100nM Staurosporine for 24 hours” and if there were ranges of concentrations or 100nM for 24 hours were the conditions.
 - o [Page 6, Western blot methods] Was the protein and RNA collected from the same samples or distinct samples?
 - o [Page 9, top] “airway cells.” Appears to be a typo or cut and paste error.

To the editorial office,

Dear Dr. Editor,

Thank you so much for the opportunity to submit a revised version of the manuscript titled "Contrasting Effects of Endogenous and Extracellular PCSK9 on Inflammation, Lipid alteration and Cell death (Submission ID: COMMSBIO-24-0829). We appreciate the comprehensive review process from the reviewers. We have revised the manuscript according to your suggestions. Please find below a point-to-point response to all the comments. Responses in Italic.

Reviewers' comments and response to the comments:

Ghalali et al sought to investigate the measurement of PCSK9 and its effect on several pathways bronchioalveolar lavage fluid (BALF). Experiments involved measurement of PCSK9 in bronchioalveolar lavage fluid (BALF), 32 single cell sequencing analysis, and exposure of lung epithelial cells to PCSK9, cigarette smoke 33 extract (CSE), and staurosporine. Their main findings are that PCSK9 stimulated cells induced proinflammatory cytokines and activation of MAPKp38.

The study might be of interested but the authors should address the following comments:

1) Despite the large body of experiments, the authors lacked of clear explanation of their results. Could you please put in context your findings of most recent evidence. For instance, you can corroborate your results discussing the study named IMPACT SIRIO published in JACC which found a remarkable antiinflammatory action of PCSK9 inhibition in severe COVID, in turn leading to signal of improved clinical events including mortality in the population with higher baseline inflammatory risk and pulmonary inflammatory disease.

Response to reviewer: *Thank you sincerely for recommending discussing the recent article highlighting the significant discovery regarding the anti-inflammatory effects of PCSK9 inhibition in severe cases of COVID. We have now discussed the findings as cited as reference 37 and incorporated it into our discussion accordingly.*

We have clarified this more in the discussion of the manuscript by adding:

"In a recent clinical study (IMPACT-SIRIO 5), patients of severe COVID-19 treated with PCSK9 inhibitor had significantly lower rates of death or need for incubation within 30 days compared to those who received the placebo. Inflammatory cytokines in serum decreased more in patients treated with the PCSK9 inhibitor compared to those on placebo. Additionally, the PCSK9 inhibitor compared to placebo, reduced mortality in patients who had higher level of baseline IL-6 levels. These findings suggest that PCSK9 regulate inflammation in lung and thus PCSK9 inhibition is beneficial in reducing inflammation and improving outcomes in severe COVID-19 cases, especially in patients with high levels of inflammation".

2) The authors' analyses should be more in-depth focused on statistical methodology to clarify significance and comparisons.

The authors never displayed numerical values and is unclear to interpret and contrast results among experiments

Response to reviewer: *Thank you for bringing up this important point. We have now conducted further statistical analyses, and to enhance clarity, we have indeed included numerical values for each statistical analysis performed.*

3) Could the authors delve into the effect on inflammation showing additional results on biomarkers of inflammation by PCSK9

Response to reviewer: For answering specifically this important question, we have conducted several experiments. We have investigated the impact of extracellular PCSK9 on caspase 1 and TGF-beta levels. We found that treatment with the extracellular PCSK9 do not affect the levels of Caspase 1 or TGF beta. This was confirmed by using several biological and technical replicates.

It is well established that caspase 1 primarily regulates IL-1 beta and the inflammasome through NFkB. This aligns with our finding (presented in the manuscript) that extracellular PCSK9 does not influence NF-kB activity. Therefore, it is rather expected that extracellular PCSK9 does not activate caspase 1. Instead, the inflammatory cytokines induced by extracellular PCSK9 appears to be regulated by alternative signaling pathways, such as MAPK p38, align with our finding (MAPK p38 activation presented in the current manuscript)

Furthermore, we conducted tests on TGF-beta, which also remained unchanged. TGF-beta plays diverse roles in homeostasis, which is also can be time and context dependent.

These new findings (Caspase 1 and TGF-beta levels) have now been incorporated to the manuscript as supplementary data/figure (Supp. Fig 1C).

4) is unclear how to interpret the effect on apoptosis induced by PCSK9. Similarly, the effect on CYP450 what clinical implications may have, you should clarify clarify this.

Response to reviewer: Our study demonstrates that in response to staurosporine, endogenous PCSK9-suppressed cells leads to higher levels of apoptosis compared to control siRNA transfected cells. Staurosporine is a protein kinase inhibitor known for its cancer cell inhibitory function. This finding suggests a potential role for endogenous PCSK9 inhibition in cancer therapy. Additionally, the diverse functions of CYP450 enzymes in liver diseases, cardiovascular diseases, and cancer underscore the importance of understanding their activities in different cancer types. Whether to suppress or enhance CYP450 activities depends on the specific cancer type and the metabolites produced by these enzymes. Variations in cytochrome P450 expression and activity can significantly influence individual responses to chemotherapy, potentially affecting treatment outcomes. Identifying the specific enzymes affected by endogenous PCSK9 could provide valuable guidance on whether suppressing PCSK9-mediated CYP450 inhibition is beneficial for cancer treatment.

These texts have been incorporated in the discussion.

This information added in the manuscript.

reviewer #2 (Remarks to the Author):

In their study, "Contrasting Effects of Endogenous and Extracellular PCSK9 on Inflammation, Lipid Alteration and Cell Death," Ghalali and colleagues focus on understanding the role and regulation of PCSK9 in the lung. PCSK9 regulates the low-density lipoprotein receptor (LDLR) and has been deployed as a lipid-lowering therapy for those at risk of cardiovascular disease. Although the liver is the major source of PCSK9, expression in other tissues such as the kidney, small intestine, brain, heart, and blood vessels suggests a role of PCSK9 beyond LDLR-mediated lipoprotein regulation (Nature Reviews Endocrinology; 2019. doi.org/10.1038/s41574-018-0110-5). As the authors nicely outline, one of the dimensions that PCSK9 could be working through is regulating inflammatory responses.

Response to reviewer: Thank you for describing the importance of our study and we appreciate for mentioning the reference. In the revised version, we have added that refence

in the introduction (ref-7).

The authors found that PCSK9 levels were higher in the BALF of smokers with or without chronic obstructive pulmonary diseases (COPD) compared to nonsmokers. PCSK9-stimulated cells induced proinflammatory cytokines and activation of MAPKp38. PCSK9 transcripts were highly expressed in healthy individuals compared to those with COPD, pulmonary fibrosis, or pulmonary systemic sclerosis. PCSK9 inhibition affected biological pathways, inducing lipid peroxidation and a higher level of apoptosis in response to staurosporine. The authors conclude that higher levels of PCSK9 in BALF act as an inflammatory marker and that extracellular and endogenous PCSK9 have different functional roles. The study has several strengths, including the significance/impact (understanding the diverse roles of PCSK9 in health and disease), the multidimensional phenotyping (human samples to cell models), and suggesting potentially distinct functional roles between intra- and extracellular PCSK9 levels. Although the significance of the study is high, a few weaknesses limit the enthusiasm for the manuscript as written:

- Please include more specifics on how subjects were identified and recruited (and the time period they were recruited over).

Response to reviewer: *We have added specific subject inclusion requirements to the material and methods section, as following: Smokers with post-bronchodilator FEV1/FVC < 0.7 and FEV1 > 40% of the predicted normal value were categorized into the COPD group, while individuals with a post-bronchodilator FEV1/FVC ratio > 0.70 and FEV1 > 80% of predicted value were classified into the non-COPD group. Never-smokers, healthy individuals with normal spirometry, were included as the control group. During the study period, all patients were clinically stable, and any individuals who had experienced airway infections within 2 weeks prior to the study were excluded. The recruitment and sample collection phase spanned a duration of 2 years.*

This information have been incorporated in the manuscript method section

- [Page 8; Figure 1F] Please clarify the source of the exogenous PCSK9 used in experimentation and if it was tested for LPS and other inflammatory impurities

Response to reviewer: *Thank you for bringing up this unintentional miss. We have now added the source of the exogenous PCSK9 and added material and methods: Recombinant PCSK9 was purchased from Sigma Aldrich (cat- SRP6285-20UG), with the source being HEK 293 cells where it was expressed. The manufacturer has reported that the endotoxin level is less than 1 EU. However, we conducted further validation to ensure the absence of lipopolysaccharide (LPS) using the Limulus Amebocyte Lysate (LAL) test. Additionally, we confirmed the absence of any impurities such as RNA or DNA by measuring spectral absorbance at 260 nm or 280 nm using a NanoDrop spectrophotometer.*

- The findings pertaining to distinct functions of intra- and extracellular PCSK9 have therapeutic implications that can be further expanded on in the discussion. Specifically, some of the newer PCSK9 RNA therapeutics could have effects on both intracellular and extracellular PCSK9 pathways compared with PCSK9 antibodies.

To discuss

Inclisiran: a small interfering RNA strategy targeting PCSK9 to treat hypercholesterolemia

Yajnavalka Banerjee 1, Anca Pantea Stoian 2, Arrigo Francesco Giuseppe Cicero 3, Federica Fogacci 3, Dragana Nikolic 4, Alexandros Sachinidis 4, Ali A Rizvi 5 6, Andrej Janez 7, Manfredi Rizzo 2 4 5

PMID: 34596005

Response to reviewer: *Thank you so much for the important suggestion. We have discussed this finding in the manuscript.*

In addition to monoclonal antibody-based inhibition of PCSK9, inhibition PCSK9 by RNA interference claimed a safer alternative for PCSK9 inhibition. RNA interference based PCSK9 inhibition has potential impact to inhibit intracellular and extracellular PCSK9. The article has been cited as reference 39.

• Minor comments:

o Consider clarifying wording surrounding “intracellular” and “endogenous” PCSK9. Endogenous can be intra or extracellular, and the authors appear to be making the distinction between intracellular and extracellular PCSK9.

Response to reviewer: *Thank you for the important suggestion! We have changed the ‘endogenous’ as ‘intracellular’ and thereby this has now been corrected.*

o As PCSK9 inhibitors in CVD have been thought to be working primarily through LDL-C regulation, in line with the author's premise, consider citing a recent PCSK9 inhibitor trial in ASCVD demonstrating modulation of inflammation in ASCVD (DOI: <https://doi.org/10.1016/j.atherosclerosis.2024.117529>).

Response to reviewer: *We have cited this recent finding (ref 38)*

o [Page 4, Cell culture methods] Please clarify: “In essence, PBEC were exposed to 100nM Staurosporine for 24 hours” and if there were ranges of concentrations or 100nM for 24 hours were the conditions.

Response to reviewer: *In a previous study, we utilized a concentration of 500nM. However, we found this concentration to be toxic, making it challenging to observe the effects of other cofactors. Hence, for this study, we optimized the assay with a reduced concentration of 100nM. This adjustment allows us to observe the effects over a period of 24 hours more clearly.*

o [Page 6, Western blot methods] Was the protein and RNA collected from the same samples or distinct samples?

Response to reviewer: *Western blot and RNA were from same samples.*

o [Page 9, top] “airway cells.” Appears to be a typo or cut and paste error.

Response to reviewer: *Thank you for pointing out the error, we have corrected this accordingly.*

REVIEWERS' COMMENTS:

Reviewer #1 (Remarks to the Author):

The authors addressed properly raised comments

Reviewer #2 (Remarks to the Author):

The authors have responded satisfactorily to my comments. I have no further suggestions.